# All Insecure, All Good? Job Insecurity Profiles in Relation to Career Correlates

**DOI:** 10.3390/ijerph16152640

**Published:** 2019-07-24

**Authors:** Nele De Cuyper, Anahí Van Hootegem, Kelly Smet, Ellen Houben, Hans De Witte

**Affiliations:** 1Research Group for Work, Organization and Personnel Psychology, KU Leuven, 3000 Leuven, Belgium; 2Optentia Research Focus Area, North-West University, Vanderbijlpark 1900, South Africa

**Keywords:** career, employability, job insecurity, Latent Profile Analysis

## Abstract

Felt job insecurity is commonly seen as a stressor that is tied to a specific segment of employees and which implies overall negative outcomes. We challenge this view based on the new career rhetoric that assumes that felt job insecurity is widespread, although not necessarily problematic; rather, on the contrary, that felt job insecurity may promote career growth and development. Accordingly, our first aim concerns the distribution of felt quantitative and qualitative job insecurity, and our second aims concerns the connection between profiles and career correlates (i.e., perceived employability, individual and organizational career management). We used two samples of Belgian employees (N1 = 2355; N2 = 3703) in view of constructive replication. We used Latent Profile Analysis to compile profiles of felt quantitative and qualitative job insecurity and linked those profiles to career outcomes. Our results are similar across samples: five profiles were found, from relatively secure to relatively insecure (aim 1). The more secure profiles reported more favorable career outcomes than the less secure profiles (aim 2). This provided overall support for the common view. We connect these findings to what we see as the main risk, namely the potentially growing divide based on felt job insecurity and the relatively large group of employees in insecure profiles.

## 1. Introduction

Felt job insecurity has been advanced as a classic work stressor in Work and Organizational Psychology [1,2] and by career scholars working within this discipline; the traditional career and standard for many employees is built on the idea of a job for life with one employer, thus job security is provided by the employer. Felt job insecurity then is a signal of precariousness and is often tied to the secondary labor market segment [3], with overall negative consequences, for example, in terms of employee health and well-being [1,2].

This view has recently been challenged in new career models: the boundaryless [4] and the protean [5] career have been advanced as contemporary alternatives when job security provided by the employer can no longer be guaranteed. These new career models have quickly found their way into the literature and have impacted the job insecurity discourse in two ways; however, so far without much attention. First, new career models are placed against the background of an increasingly VUCA world. VUCA is a popular acronym to describe heightened levels of volatility, uncertainty, complexity and ambiguity in the labor market, affecting both organizations and the individuals working in those organizations. In concert, those factors are variations on two underlying dimensions: unpredictability and uncontrollability [6], both at the core of felt job insecurity [1,7]. VUCA and similar catch-all phrases have strengthened the idea that all employees feel insecure, be it about the future of their jobs (i.e., felt quantitative job insecurity) or specific aspects of the job (i.e., felt qualitative job insecurity). Hence, felt job insecurity is no longer considered a signal of precariousness for a specific contingent of workers. Secondly, and perhaps even more fundamentally, the new career rhetoric assumes that felt job insecurity can be an impulse for growth and development [8]; it puts much emphasis on individual agency. Felt job insecurity is seen as a trigger for workers to exert agency by exploring alternatives and engaging in career management. This is in contrast with the more traditional view that felt job insecurity leaves scars; felt job insecurity negatively affects health, well-being and employee attitudes and likely also the future career [2]. To date, felt job insecurity in relation to career outcomes has not yet been probed in much detail [9].

We set out to challenge the traditional view on felt job insecurity along the new career rhetoric. This challenge involves two related aims. Our first aim is to probe how felt job insecurity is distributed. This is done to challenge the traditional idea that felt job insecurity is distributed unequally, in response to the idea that felt job insecurity affects an increasingly larger contingent, if not all workers. This challenge is important, as it could mean that employees have embraced felt job insecurity as ‘the new normal’, which in popular writings seems to have become quite mainstream. We achieve this aim by probing how felt quantitative and qualitative job insecurity are combined within employees using latent profile analysis. Felt quantitative job insecurity relates to concerns about potential job loss, and felt qualitative job insecurity to concerns about how the job will look like in the future [10,11]. To date, qualitative job insecurity has attracted comparatively little research attention [12], and the combination of felt quantitative and qualitative job insecurity is rarer still (see [13,14] for exceptions). Latent profile analysis is still relatively new in a domain that is dominated by a variable-centered approach (see [15] for exceptions). A within-person approach is critical, though, as it could illustrate heterogeneity within the workforce that would otherwise go unnoticed.

Our second aim is to link those profiles to career-relevant aspects. This is done to provoke the dominant idea that felt job insecurity is all bad; an idea that may need rethinking in a VUCA era according to popular writings. This is important to establish whether felt job insecurity is a hindrance, as it is commonly perceived, or rather a challenge [16]. We focus upon perceived employability and career management as important career correlates. Perceived employability concerns the individual’s perception of chances in the internal (i.e., perceived internal employability) or external (i.e., perceived external employability) labor market. It has particular resonance in this study as the assumed new security [17]. Career management refers to all individual (i.e., individual career management) or organizational (i.e., organizational career management) practices that attempt to influence the individual’s career development [18,19]: individual and organizational career management are both critical and complementary features in new career models [8]. We will focus upon networking behavior as an indicator of individual career management: networking behavior has particular value, both instrumental and emotional, when employees feel insecure [8], and is commonly included in measures tapping into individual career management [19,20,21]. We will focus upon employees’ perceptions of formal (e.g., providing training) and informal (e.g., providing career advise) developmental and supportive practices as indicators of organizational career management: these are at the core of organizational career management [22] and are generally seen as the alternative for earlier top-down career practices [20].

We use two samples of Belgian workers. Most respondents from Sample 1 are highly educated, with a relatively large share in management positions: those respondents are likely more agentic, also in the sense of being able to cope with feelings of felt job insecurity. This sample serves to challenge the traditional view on felt job insecurity. In contrast, Sample 2 is more heterogeneous in terms of both education and position; this sample allows testing whether the pattern found in Sample 1, in terms of both distribution (aim 1) and career correlates (aim 2), can be generalized to a more diverse population.

### 1.1. Profiles of Felt Job Insecurity

Felt job insecurity is a ‘perceived threat to the continuity and stability of employment as it is currently experienced’ [7] (p. 1914); at the heart of this definition is that felt job insecurity is perceptual, future oriented and related to the current job in the current organization. The current job may concern the job as a whole or features of the job, labeled felt quantitative and qualitative job insecurity, respectively [10,11]. We are interested in how felt quantitative and qualitative job insecurity are combined within profiles of employees and how they are distributed across those profiles. Profiles are described in terms of level and shape [23]. Level describes profiles in terms of how insecure they feel, from low to high. Shape describes profiles in terms of different forms.

Level refers to differences in how insecure employees from different profiles feel, from relatively secure to insecure. When thinking in line with the traditional view on careers, we expect a strong divide between secure profiles and increasingly more insecure profiles. More specifically, we expect a relatively large group of highly secure profiles and then layers of increasingly insecure profiles: this fits the idea of a core-periphery structure that has been highlighted in the context of labor market segmentation and applied in work and organizational psychology [24]. In contrast, boundaries between core and periphery are blurred in the new career discourse [25], and so may be the divide between secure and insecure profiles.

Shape refers to how felt quantitative and qualitative job insecurity combine within profiles. Thinking along the frameworks used in support of the dominant idea, felt quantitative and qualitative job insecurity are likely strongly interwoven. For example, Conservation of Resources Theory [26] highlights that resources—and hence also lack of resources, in the form of insecurity—often go together. Similarly, studies on labor market segmentation assume and demonstrate that jobs in the secondary segment are more precarious than jobs in the primary segment on many different indicators, including felt quantitative and qualitative job insecurity [27]. This idea is challenged when profiles take different shapes, with different possible combinations of felt quantitative and qualitative job insecurity as a result of ongoing labor market dynamics. For example, employees may feel insecure about their job because of increased contractual flexibility, with consequences for personnel staffing [28], and the many restructurings announced in media. Likewise, they may feel insecure about how their job will look like because jobs are seen as dynamic rather than static and are tied to specific and changing roles. The connection between changing roles and felt qualitative but not necessarily quantitative job insecurity has been described against the background of the many changes in the public sector (see [29] for an example); employees in the public sector traditionally feel secure about their job, but not necessarily about how their job will look like in the future. Empirical evidence from variable-centered studies suggests a positive correlation between quantitative and qualitative job insecurity. Yet, correlations vary from low (e.g., *r* = 0.14 in [30]), over medium (e.g., *r* = 0.26 in [13]) to high (e.g., *r* = 0.82 in [31]; *r* = 0.79 in [32]). 

In short, the traditional view is challenged when there is a blurred distinction between secure and insecure profiles and when there is variation in shapes. We do acknowledge that the distinction between profiles in terms of both level and shape is relative and conditional upon interpretation.

### 1.2. Career Correlates

In a next step, we relate profiles to perceived employability, individual and organizational career management. Career correlates of specific profiles can help to add theoretical meaning [23] and possibly substantiate one of the views.

Felt job insecurity in the traditional view on careers and dominant theoretical frameworks within work and organizational psychology is advanced as a stressor [1,2] and a signal of precariousness [3], which could spill over to matters related to the career [33]. This ties in with insights from Conservation of Resources theory [26], namely that those with fewer resources are prone to ongoing resource loss and vice versa for those with many resources. This is likely the result of two interrelated dynamics of career management. First, felt job insecurity hampers individual career management. Employees who feel insecure tend to become risk-averse: they try to defend what they perceive to have in their current job [17]. This defensive reaction implies that employees refrain from making investments that require energy, but at the same time such reaction consumes energy. The inaction associated with defense obviously forestalls career development [34]. In contrast, employees who feel secure have spare resources to invest in their career. Second, felt job insecurity reduces organizational career management. Security gives access to the collective pool of resources available in the organization; this complies with the observation that organizations tend to invest more in those on jobs that contribute to the core of organizational functioning and less in those at the periphery with workers who are easily replaceable [24]. The result of these two dynamics is that employees who feel insecure may also feel less employable in both the external and the internal labor market. Their natural defensive reaction and unequal access to organizational resources narrows down the perceived window of available employment opportunities in the internal and external labor market [17,35].

This view is challenged when felt job insecurity is the new reality for employees [25]; employees comply with and adapt to increasing insecurity, simply because they have to [33,36]. A career for life within one organization is no longer guaranteed or perhaps desired, which induces a pressure of mobility or in any case proactive coping [7]; employees seek ways to cope ahead of time with job or job feature loss. This involves engaging in career management, individual as well as organizational, and gauging the internal and external labor market [8]. Taking this one step further, a plausible assumption could be that lack of necessity, when employees feel secure, leads to complacency [7].

The evidence from variable-centered studies is growing, but so far is mixed for perceived employability, and absent for career management. Most studies hint at a negative relationship between felt quantitative job insecurity and perceived, mostly external, employability, though with substantial variation in strength: from strong (*r* = −0.58 in [36]), through moderate (*r* = −0.37 in [30]; *r* = −0.29 [37]) to weak (*r* = −0.14 [38]; *r* = −0.16 in [39]). The studies by Wang et al. [39] and Kang et al. [30] suggest similar negative relationships between felt qualitative job insecurity and perceived external employability. However, other studies do not establish a significant relationship or report a positive association. For example, Sora et al. [40] report a null correlation and Fontinha et al. [41] report a positive correlation between felt quantitative job insecurity and perceived external employability. What this second stream of studies have in common is the sample of employees in Southern countries where the recent crisis has struck particularly hard, so that felt job insecurity has perhaps been accepted as part of everyday life to which people have adapted [42]. A plausible assumption is that similar dynamics may be at play within profiles: when felt job insecurity is widespread and high, employees may have adapted and have embraced employability as the new security mechanism.

In all, we expect the most insecure profiles to report poorer career outcomes, i.e., lower perceived employability, and lower engagement in individual and organizational career management when thinking along the traditional view. This is challenged by the idea, when stretched to its extreme, that the most insecure profiles are pushed towards action and reflection about their career, while job secure profiles settle into inaction.

## 2. Methods

### 2.1. Participants and Procedure

#### 2.1.1. Sample 1

Data were collected among 2355 Flemish (i.e., the Dutch-speaking region of Belgium with approximately six million inhabitants) readers from an online HR magazine in 2017. They responded to a call published on the magazine’s website and in the newspaper to participate in a survey on occupational health and well-being, with a link to the survey. To increase the response rate, gift vouchers were raffled among employees who participated in the survey. All respondents received an online informed consent form, which clearly stated that participation was voluntary and that data will be processed anonymously.

Slightly more women (58.8%) than men (41.2%) participated. Average age was 40.92 (*SD* = 10.59), and average job tenure was 10.52 years (*SD* = 9.45). The large majority had a degree of higher education (75.0%), worked on a full-time (80.5%) and permanent (95.1%) contract in the private sector (80.9%). A minority worked as blue-collar workers (5.9%) or had a top management position (8.5%). Most respondents were lower- (26.7%) or higher- (34.8%) level white-collar workers, or held a lower management position (24.1%).

#### 2.1.2. Sample 2

We launched a call for participation in a study on employability during a series of presentations for human resource practitioners in Flanders. Human resource practitioners from 14 organizations volunteered to participate: five organizations from the service industry, three organizations from the public sector, two organizations from the social economy, two organizations in higher education and two organizations from the manufacturing industry. Participation was awarded with a feedback report.

Data was collected between January and March 2013 and was facilitated by the human resource practitioners who provided access to employees. The research team sent out 8180 invitations for participation, along with a link to the survey or paper and pencil questionnaires, when preferred by the respondents. Participation was encouraged by raffling prizes. Participation was voluntary and based on informed consent, and full confidentiality was guaranteed. Surveys were filled out at work or at home, as chosen by the respondents. Three thousand seven hundred and three respondents participated in the survey.

Slightly more men (55%) than women participated. Average age was 41.85 (*SD* = 10.46) and respondents had, on average, been employed for 12 years (*SD* = 10.59) in their current organization. A total of 54.3 per cent of the respondents had a degree of higher education. The large majority worked on a full-time basis (80.4%) and had a permanent contract (90.4%). About 25 per cent of the sample were blue collar workers, either unskilled (7.8%) or skilled (17.8%). Other respondents were lower white-collar workers (17.6%) or higher white-collar workers (18.8%), or held a lower (28.8%) or top (7.9%) management position.

### 2.2. Measures

Means, standard deviations, and correlations among the variables are presented in Table 1 and Table 2 for Samples 1 and 2, respectively. All variables were assessed using a five-point Likert-type scale ranging from 1 (totally disagree) to 5 (totally agree).

Felt quantitative job insecurity (profile indicator). Felt quantitative job insecurity was measured with the four-item Job Insecurity Scale, developed by De Witte [43] and validated by Vander Elst, De Witte, and De Cuyper [44]. A sample item is “I think I will lose my job in the near future”. Cronbach’s alpha was 0.928 for Sample 1 and 0.853 for Sample 2.

Felt qualitative job insecurity (profile indicator). Felt qualitative job insecurity was measured with four items from De Witte, De Cuyper, Handaja, Sverke, Näswall, and Hellgren [45]. A sample item is “I am worried about how my job will look like in the future”. Cronbach’s alpha was 0.902 in Sample 1 and 0.912 in Sample 2.

Perceived employability (outcome). In Sample 1, perceived employability was measured with the four items used in [46,47]. A sample item is “I could easily find another job, if I wanted to”. Cronbach’s alpha was 0.965. In Sample 2, perceived employability was measured with eight items from De Cuyper and De Witte [48]. Four items related to perceived internal employability, for example “I can easily find another job in this organization, instead of my present job”. Cronbach’s alpha was 0.907. The other four items related to perceived external employability, for example “I can easily find another job with another employer, instead of my present job”. Cronbach’s alpha was 0.941.

Organizational career management (outcome). Organizational career management was measured in Sample 2 using the instrument developed by Sturges et al. [19]. Six items relate to formal career management practices, for example “I have been given training to help develop my career”. Four items relate to informal career management practices, for example “I have been introduced to people at work who are prepared to help me develop my career”. Cronbach’s alpha was α = 0.852 for formal career management practices and 0.879 for informal career management practices.

Individual career management—networking behavior (outcome). Networking behavior was measured in Sample 2 with seven items developed by Sturges et al. [19]. A sample item is: “I have got myself introduced to people who can influence my career”. One item (“I have refused to accept a new role because it would not help me develop new skills”) was excluded as the standardized factor loading estimate was lower than 0.50. Cronbach’s alpha, based on the six included items, is 0.850.

### 2.3. Preliminary Analysis

Mixture models with latent scores (i.e., items are used to estimate latent factors) control for measurement error and are therefore preferred. Yet, the complexity of those models often leads to the convergence of statistically unsound models or no convergence at all [49], as in our study. Factor scores have been advanced as an alternative: they partially control for measurement error by assigning more weight to items with lower levels of measurement error [50]. Moreover, they allow evaluation measurement invariance across groups (in this study: samples), and to preserve this measurement structure by saving the factor scores from the most invariant measurement model [51].

Accordingly, we first assessed the measurement invariance of these constructs (results available upon request from the first author) (see [51] for a similar approach). We first compared an unconstrained model to a metric invariance model in which factor loadings were equal across samples. Subsequently, we compared this model to a scalar invariance model in which factor loadings and intercepts were constrained to be equal across samples. We obtained scalar invariance for felt qualitative job insecurity and partial scalar invariance for felt quantitative job insecurity after freeing one item (namely, the reverse-scored item) of the felt quantitative job insecurity scale. The implication is that comparisons regarding the means of felt quantitative job insecurity across samples should be made with caution. The factor scores from this model were saved and used for further analyses.

### 2.4. Latent Profile Analysis

We conducted latent profile analysis (LPA) to identify profiles of felt quantitative and qualitative job insecurity by means of Mplus version 8. LPA is a method based on continuous variables which analyses data from heterogeneous populations and models it into clusters of participants in the sample [52]. All analyses were conducted using the robust maximum likelihood estimator (MLR), 5000 random sets of start values, and 1000 iterations. We first conducted LPA in Sample 1. The invariant factor scores of felt quantitative and qualitative job insecurity were simultaneously entered. Subsequently, we conducted LPA with the invariant factor scores derived for Sample 2 in view of replication.

In line with the recommendations by Nylund, Asparouhov, and Muthén [53], we employed several goodness-of-fit indicators. First, we used the Bayesian information criterion (BIC), which makes it possible to compare models with varying numbers of classes while adjusting for the number or parameters in the model and sample size [54]. Lower values indicate a better model fit. Furthermore, we used the Lo–Mendell–Rubin likelihood ratio test (LMR) and bootstrap likelihood ratio test (BLRT) to evaluate model fit. LMR approximates the distribution of the likelihood ratio difference test, and BLRT uses bootstrap samples to empirically estimate the difference distribution [53]. Both tests provide a *p*-value that indicates whether there is an improvement in model fit when adding an extra profile: a significant *p*-value indicates that a model with k + 1 classes significantly outperforms a model with k classes. Please note that we did not use the Akaike information criterion (AIC) or sample size-adjusted Bayesian information criterion (SSBIC): AIC has a tendency to overestimate the correct number of classes [53], and SSBIC is less likely to identify a correct solution in models with more continuous than categorical indicators [54].

Finally, we evaluated the entropy and the smallest profile size to verify model fit. The entropy indicates the degree of certainty that an individual is assigned to a correct class [55], in which higher values indicate clearer class separation (ranging from 0 to 1) [56]. Entropy values of 0.40, 0.60 and 0.80 are considered to represent low, medium and high class separation, respectively [57]. The smallest profile should contain at least 1% of the sample and/or the n should consist of a minimum of 25 respondents, as smaller classes might increase the possibility of low power and precision [58].

### 2.5. Outcomes of Profile Membership

We probed whether the profiles in Sample 1 differed with respect to perceived employability, and in Sample 2 with respect to perceived internal and external employability, individual career management in the form of networking behavior, and organizational career management in the form of formal and informal developmental and supportive practices. We used the Auxiliary (BCH) command in Mplus [59,60]. This provides an overall test and pairwise comparisons of the between profile means using Wald chi-square tests while maintaining the initial profile solution [61]. We also ran these analyses including educational level as a covariate, by means of the manual BCH estimation. This method avoids class shifts, and allows controlling for covariates when relating latent classes to a distal outcome [59]. Since analyses with or without educational level generated similar results, we proceeded with the most parsimonious model (i.e., no control variables) [62].

## 3. Results

### 3.1. Profiles of Felt Quantitative and Qualitative Job Insecurity

Latent profile analyses were conducted for one- to five-class solutions in Samples 1 and 2. The five-profile solution provided the best fit to the data in both samples. Details related to the selection of the optimal profile are shown in Appendix A. Graphical representations of the profiles, along with factor and scale (between brackets) scores, are shown in Figure 1 and Figure 2. Factor scores are relative to the sample’s average; they show whether a specific profile feels more or less insecure than the average in the sample. Scale scores are absolute (see our interpretation below).

#### 3.1.1. Sample 1

The level of both felt quantitative and qualitative job insecurity increased from profile 1 to profile 5 (Figure 1). By way of illustration, employees in profile 1 felt secure relative to the sample’s average (factor score felt quantitative job insecurity: −1.26; factor score felt qualitative job insecurity: −0.65), and in absolute terms (scale score for felt quantitative job insecurity: 1.21, for felt qualitative job insecurity: 2.49). Employees in profile 5 felt relatively insecure, both quantitatively (factor score: 2.37) and qualitatively (factor score: 0.93), and in absolute terms, both quantitatively (scale score: 4.19) and qualitatively (scale score: 3.81).

All profiles were similar in shape, with felt quantitative and qualitative job insecurity covarying. Please note that scale scores, unlike factor scores, were higher for felt qualitative than for felt quantitative job insecurity for profiles 1 to 3, but not profiles 4 and 5.

In terms of profile size, profile 2 included most respondents (37.45%; *N* = 882). Profiles 1 (22.34%; *N* = 527) and 5 (24.29%; *N* = 572) were similar in size. Profiles 3 (10.96%; *N* = 258) and 4 (4,93%; *N* = 116) were comparatively small.

#### 3.1.2. Sample 2

Profiles were virtually the same in Sample 2 (Figure 2). The level of felt job insecurity increased from profile 1 to profile 5, with the exception of felt quantitative job insecurity in profiles 2 and 3; felt quantitative job insecurity was lower in profile 3 (factor score: −1.01; scale score: 1.54) than profile 2 (factor score: −0.36; scale score: 2.28). Felt quantitative and qualitative job insecurity covaried. Scale scores showed higher felt qualitative than quantitative job insecurity for all profiles, except profile 5. Profiles 1 (36.92%; *N* = 1367) and 2 (38.40%; *N* = 1422) included most respondents, followed by profile 4 (17.26%; *N* = 639). Profiles 3 (5.67%; *N* = 210) and 5 (1.76%; *N* = 65) were comparatively small.

### 3.2. Differences between Profiles in Terms of Career Correlates

#### 3.2.1. Sample 1

We compared the profiles in terms of perceived employability. Respondents from profile 1 (*M* = 0.421, *SD* = 0.046) felt most employable and more employable than the other profiles (Table 3). Respondents from profile 2 (*M* = 0.081, *SD* = 0.035) were second highest and were significantly higher than profiles 3, 4, and 5. Respondents from profiles 4 (*M* = −0.439, *SD* = 0.069) and 5 (*M* = −0.436, *SD* = 0.121) did not differ significantly (*p* = 0.984) and felt least employable and less employable than respondents from profile 3 (*M* = −0.227, *SD* = 0.045). Please note that respondents from profiles 3 and 5 did not differ significantly (*p* = 0.104).

#### 3.2.2. Sample 2

The following outcomes were used: perceived internal and external employment, individual career management in the form of networking behavior and organizational career management in the form of formal and informal developmental and supportive practices (Table 3).

In general, the pattern was similar across the outcomes: respondents from profile 1 were highest, followed by respondents from profile 2. Respondents from profile 3 were lowest and from profile 5 s lowest. Profile 4 was situated in between. Differences between profiles were significant, unless described otherwise in the following.

For perceived internal employability, profiles 3 (*M* = −0.364, *SD* = 0.087) and 5 (*M* = −0.313, *SD* = 0.073, *p* = 0.652) did not differ significantly. For perceived external employability, profile 5 (*M* = −0.069, *SD* = 0.079) did not differ significantly from the other profiles (profile 1: *M* = 0.062, *SD* = 0.031, *p* = 0.121; profile 2: *M* = 0.046, *SD* = 0.025, *p* = 0.163; profile 3: *M* = −0.286, *SD* = 0.099, *p* = 0.085; profile 4: *M* = −0.078, *SD* = 0.037, *p* = 0.915). Profiles 1 and 2 did not differ significantly either (*p* = 0.693). For networking behavior from the category individual career management, profile 5 (*M* = −0.088, *SD* = 0.067) did not differ significantly from profiles 2 (*M* = 0.002, *SD* = 0.021, *p* = 0.202), 3 (*M* = −0.274, *SD* = 0.074, *p* = 0.063) and 4 (*M* = −0.069, *SD* = 0.030, *p* = 0.807). Differences between profiles 2 and 4 were marginally significant (*p* = 0.058). For formal career management practices, profiles 3 (*M* = −0.528, *SD* = 0.085) and 5 (*M* = −0.375, *SD* = 0.073, *p* = 0.174) did not differ significantly. For informal career management practices, all differences were significant.

## 4. Discussion

The aim of our study was to challenge the dominant idea that felt job insecurity is a signal of precariousness [3], and hence tied to specific groups and related to overall negative consequences. This challenge was inspired by assumptions from the new career rhetoric that felt job insecurity has become widespread and could leverage growth and development. We achieved this aim by testing felt job insecurity profiles and career correlates in two samples.

The first aim concerned the distribution of felt quantitative and qualitative job insecurity in profiles. We identified five profiles in Samples 1 and 2: employees from profile 1 felt least insecure and employees from profile 5 felt most insecure, with layers of decreasing insecurity between those profiles. Felt quantitative and qualitative job insecurity covaried. The only exception was found in Sample 2: employees from profile 2 felt more quantitatively insecure than employees from profile 3. The relatively secure profiles 1 and 2 form the majority in Sample 1 (59.80%) and the large majority in Sample 2 (75.32%). The relatively insecure profiles 4 and 5 form a significant minority, 29.22% and 19.02% in Samples 1 and 2 respectively. These results align with the traditional idea: there still is a fairly strong divide, with a majority of employees feeling secure. This evidence seems to put the new career rhetoric in perspective. This is in alignment with the work from Biemann, Fasang and Grunow [63] and Hollister [64], who argue that changes in career patterns are overestimated.

The second aim concerned the association between felt job insecurity profiles and career correlates: perceived employability, individual career management in the form of networking behavior, and organizational career management in the form of formal and informal developmental and supportive practices. The focus upon careers is relatively new in job insecurity research. The results from Sample 1 are straightforward: employees in more secure profiles felt more employable than those in less secure profiles. The results from Sample 2 are similar: perceived internal and external employability, individual and organizational career management are highest in profiles 1 and then 2, followed by profiles 4 and 5, and lowest in profile 3. Though not all comparisons are significant, the overall picture supports the traditional view. Secure profiles have positive career correlates: employees in those profiles perceive more chances in the labor market, act upon their career, and receive support and opportunities for development from their organization. In contrast, insecure profiles have negative career correlates, suggesting that felt insecurity also signals precariousness in career-related matters.

This pattern of results highlights a potential risk: those in the most insecure profiles do not engage in career management. This apparent inaction among the most insecure profiles may be paradoxical as they are most in need for action (see [35] for a similar discussion). The underlying dynamics have been described in different theoretical frameworks. For example, Conservation of Resources theory [26] highlights that those with many resources are prone to collect more resources, and vice versa for those with fewer resources. Another example concerns recent insights about career inaction [34]: individuals often do not act when outcomes are uncertain. Felt job insecurity presents a double vulnerability, namely, outcome uncertainty in the present and the future job; employees do not know what will happen in their present job and they feel less employable for a future job.

The pattern of results is not entirely clear-cut: employees in profile 3 reported the poorest career outcomes in Sample 2. Profile 3 in Sample 2 was fairly small with only 5.67% of the respondents, and fairly specific: they feel relatively secure about their job and *more* secure than employees in Profile 2 and relatively insecure about how their job will look like in the future. Though interpretations are somewhat tentative, this seems to suggest that felt qualitative job insecurity is particularly important in view of career correlates.

In all, it seems safe to conclude that the overall pattern of results supports the view that is still common in job insecurity research; a view that is based on the traditional career. First, a (large) majority feels secure about their jobs, both quantitatively and qualitatively, but with a significant minority feeling insecure. Second, the least secure profiles report to have fewer opportunities in the labor market and are less engaged in career management. This is in contrast with ideas related to the new career discourse; there, the general idea is that external pressures will lead to volition career management, which is clearly not what we found. Yet, results are not entirely straightforward: we identify several routes for research below.

### Strengths and Limitations

Our paper has several strong features. First, we included felt quantitative and qualitative job insecurity as indicators of an overall insecurity profile. Previous studies have treated felt quantitative and qualitative job insecurity as separate constructs because they have different foci [11,13,14]; the corresponding research question then concerns which type of felt job insecurity is most problematic in terms of outcomes. Our study shows that felt quantitative and qualitative job insecurity are likely dependent: it seems that felt quantitative job insecurity implies qualitative job insecurity for many workers. Those conceptual issues have not been debated thoroughly: our results may serve this debate.

Second, we used two samples in view of challenging the dominant thesis and constructive replication. The samples used in this study are clearly different in terms of data collection, timing and respondents’ backgrounds. Sample 1 was collected in 2017 using an open invitation targeted at individuals with an overall strong educational background who are relatively high on the hierarchical ladder. This sample is particularly well suited to challenging ideas associated with the traditional career; respondents in this sample are well equipped to cope with felt job insecurity. Sample 2 was collected in 2013 via organizations, with the explicit aim of heterogeneity and in view of constructive replication. We established similar results across these samples, which strengthens our conclusions and hint at possibilities for generalization. In contrast to Sample 1, it could be interesting to oversample workers who are employed on temporary or otherwise flexible contracts. Eurostat estimates the share of temporary workers in Belgium at 7%. This is similar to the shares in the samples but perhaps too limited to provide a view on potential career risks for those in the secondary labor market segment.

Third, we used a within-person approach. This approach provides new information compared to the more common between-person approach. For example, it is often said that felt job insecurity *on average* is increasing [2], yet this may mask the large heterogeneity within the workforce. A plausible assumption is that felt insecurity is particularly high among some workers, as we showed in our study.

That said, we acknowledge that our study comes with some limitations. First, data are cross-sectional with obvious disadvantages compared to longitudinal data. A first disadvantage is that we cannot argue that specific profiles *cause* poorer career outcomes. Reversed causation, so that those who feel less employable or invest less in their career subsequently feel more insecure, is plausible. Nevertheless, we do think felt job insecurity is more likely to affect the career than vice versa: the reason is that felt job insecurity concerns the present job in the current organization, and the career relates to the future. It seems reasonable to assume that the present situation forms the basis for future outcomes. A second disadvantage is that we cannot follow specific profiles over time. This could present a route for future research, also in view of testing a potential Matthew effect; this would occur if the divide between secure and insecure profiles were to be replicated or become even larger over time. It would be a challenging route, though, as Kinnunen et al. [15]) demonstrated that felt job insecurity is relatively stable over time, even in a context of ongoing change.

Second, we used convenience samples, and not samples that were representative of the Belgian population. Though we are not so much concerned about the level and shape of the specific profiles given replication across the samples, it may have affected the size of specific profiles; indeed, the size, more than level or shape, varied across the samples. An important route for future research concerns the use of representative samples; this would allow making more accurate estimations about the size of the different profiles and it could provide opportunities to identify who is at risk by linking profiles to socio-economic background.

Third, we used an overall and generic measure for felt qualitative job insecurity, while most other measures probe into specific aspects of the job, for example worries about pay or career prospects [11]. The current state of the art is more advanced for felt quantitative than for qualitative job insecurity, both in terms of number of studies, conceptual refinement and measures. For example, Schoss [7] convincingly argues that felt quantitative job insecurity should be probed through overall perceptions of job loss rather than through multiplicative scales. This has become the norm in the field, with scales from De Witte [43,44], Probst [65] and Hellgren et al. [11]. Yet, a similar discussion is lacking for felt qualitative job insecurity.

Fourth, we are very well aware that perceived employability, individual and organizational career management are a specific selection of career outcomes; they serve as a first step in bringing matters related to the career into job insecurity research. Future research may gradually strengthen and broaden this selection. Strengthening may take the form of focusing upon bundles of individual and organizational career management practices (see e.g., [20]), rather than isolated practices. Broadening implies including other career outcomes, for example career success.

## 5. Conclusions

The idea that felt job insecurity is widespread, though not necessarily bad for the career, is quickly gaining momentum in the psychological career literature; felt job insecurity is sometimes portrayed as the new normal for all workers and as a trigger for career development. Our results do not comply with this view; rather, they suggest the contrary. There are different insecurity profiles, from relatively secure to insecure, and they go together with career-related matters; employees in the most secure profiles feel more employable with a stronger engagement in individual and organizational career management, and vice versa for the least secure profiles. This is a cause for concern, as it could imply that the divide in security may be replicated in other fields with lasting scars. This concern is pressing given the size of the insecure profiles.

## Figures and Tables

**Figure 1 ijerph-16-02640-f001:**
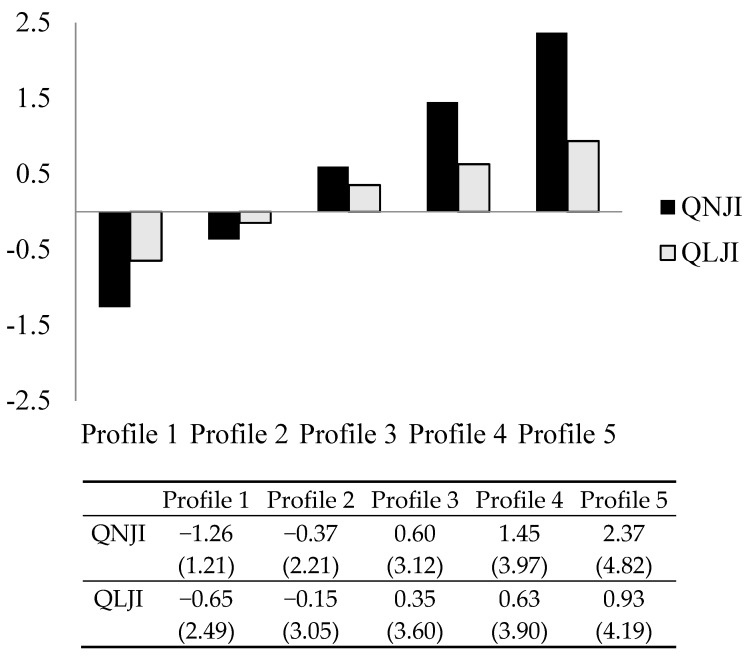
Graphical representation of the profiles of Sample 1. Scale scores between brackets. Q(N/L)JI = Felt Quantitative/Qualitative Job Insecurity.

**Figure 2 ijerph-16-02640-f002:**
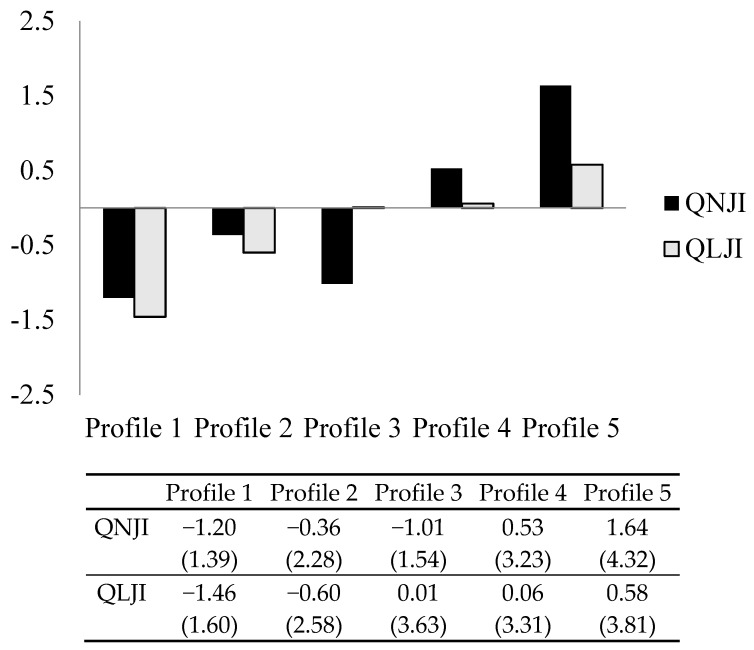
Graphical representation of the profiles of Sample 2. Scale scores between brackets. Q(N/L)JI = Felt Quantitative/Qualitative Job Insecurity.

**Table 1 ijerph-16-02640-t001:** Means, standard deviations and correlations for the study variables of Sample 1.

Variables	Means	*SD*	1	2	3
1. Felt quantitative job insecurity	2.50	1.04	(0.93)		
2. Felt qualitative job insecurity	3.21	0.97	0.52 **	(0.90)	
3. Perceived employability	3.35	1.07	−0.27 **	−0.29 **	(0.97)

Note. ** *p* < 0.01; Cronbach’s alpha on the diagonal.

**Table 2 ijerph-16-02640-t002:** Means, standard deviations and correlations for the study variables of Sample 2.

Variables	Means	*SD*	1	2	3	4	5	6	7
1. Felt quantitative job insecurity	2.22	0.87	(0.85)						
2. Felt qualitative job insecurity	2.52	1.03	0.54 **	(0.91)					
3. Perceived internal employability	2.54	0.88	−0.11 **	−0.16 **	(0.91)				
4. Perceived external employability	3.12	0.98	−0.05 **	−0.05 **	0.28 **	(0.94)			
5. Formal CM practices	3.14	0.80	−0.14 **	−0.29 **	0.27 **	0.04 *	(0.85)		
6. Informal CM practices	2.36	0.94	−0.09 **	−0.26 **	0.30 **	0.06 **	0.66 **	(0.88)	
7. Networking	2.81	0.78	−0.07 **	−0.11 **	0.23 **	0.23 **	0.31 **	0.36 **	-

Note. * *p* < 0.05, ** *p* < 0.01; Cronbach’s alpha on the diagonal; CM = Career Management.

**Table 3 ijerph-16-02640-t003:** Equality tests of means (SD) across the profiles using the BCH procedure.

Study/Outcome	Profile 1 (A)	Profile 2 (B)	Profile 3 (C)	Profile 4 (D)	Profile 5 (E)	Chi-Square Overall Test (*df* = 4)
**Sample 1**						
Perceived employability	0.42 (0.05) ^BCDE^	0.08 (0.04) ^ACDE^	−0.23 (0.12) ^ABD^	−0.44 (0.07) ^ABC^	−0.44 (0.05) ^AB^	176.11 **
**Sample 2**						
Perceived internal employability	0.14 (0.03) ^BCDE^	0.03 (0.02) ^ACDE^	−0.36 (0.09) ^ABD^	−0.09 (0.03) ^ABCE^	−0.31 (0.07) ^ABD^	61.51 **
Perceived external employability	0.06 (0.03) ^CD^	0.05 (0.03) ^CD^	−0.29 (0.10) ^ABD^	−0.08 (0.04) ^ABC^	−0.07 (0.08)	17.74 **
OCM Formal	0.21 (0.03) ^BCDE^	0.03 (0.02) ^ACDE^	−0.53 (0.09) ^ABD^	−0.12 (0.03) ^ABCE^	−0.38 (0.07) ^ABD^	124.65 **
OCM Informal	0.14 (0.03) ^BCDE^	0.03 (0.02) ^ACDE^	−0.47 (0.07) ^ABDE^	−0.08 (0.03) ^ABCE^	−0.25 (0.06) ^ABCD^	92.69 **
ICM Networking	0.11 (0.03) ^BCDE^	0.00 (0.02) ^AC^	−0.27 (0.07) ^ABD^	−0.07 (0.03) ^AC^	−0.09 (0.07) ^A^	34.72 **

Note. The values for each variable are standardized means per profile. Superscripts indicate profiles that are significantly different at ** *p* < 0.05; OCM = organizational career management; ICM = individual career management.

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
