# Peer review of "All Insecure, All Good? Job Insecurity Profiles in Relation to Career Correlates"

_ijerph, 2019, doi:10.3390/ijerph16152640_

Round 1

Reviewer 1 Report

Thank you for this paper. I was very much interested in the new findings in how to theorize and measure job insecurity in terms of the effects on careers. The paper fits well with the Special Issue on "The Impact of Job insecurity on Non-Traditional Outcomes", which has a focus on work psychology and related theoretical discussions.

The manuscript is clearly written and coherent overall. My concern relates to the framing of the paper and to the use of labour market segmentation theory as well as to the concept "career". This is not to reject how authors operationalise etc., but to remind about the very different uses of the concepts "labor market", and "career". In my social science and labour market research background, the labour market and "careers" refer to evaluating the socio-economic position of the workforce and worker transitions and mobility at the labour market (i.e. transitions from employment to unemployment, occupation & job changes, salary progression, etc.), by worker/firm/region/country characteristics and antecedents. I am sure the authors know these differentiating discussions; I am asking to revise the text keeping in mind that "career literature", the term used e.g. on p.2, row 21, can also be something quite different. I would rather refer to “work psychology career literature”.

Following that, I agree that the main idea of labour market segmentation theory incl. division between core and periphery labour market segments, with the periphery segment suffering from increased insecurity, is related. However, I am not sure if the theory has been used in this context. I would rather discuss this theory as part of the general framework and discussion and by stating how the findings can contribute. This is because the paper does not evaluate labour market segments at all by socio-economic background, gender, skills, contractual issues, etc. Another thing is that the paper seems to have quite selected, perhaps "best off" segments of Belgian labour force, with with 95% and 90% of participants having permanent contracts in samples 1 and 2, respectively. How do these samples represent labour market segments in Belgium?

Please keep in mind also that labour market oriented career research does not find any overall deterioration of careers, and it has been debated if there ever existed any “golden era” when everything was “stable”. Please consider, in this sense, your use of concept “new career perspective” – there are myths related. You can have a look at e.g.:

Biemann T., Fasang A.E. & Grunow D. (2011) Do economic globalization and industry growth destabilize careers? An analysis of career complexity and career patterns over time. Organization Studies 32 (12): 1639–1663.

Hollister M. (2011) Employment stability in the U.S. labor market: Rhetoric versus reality. Annual Review of Sociology 37(1): 305–324.

Van Winkle Z. & Fasang A. (2017) Complexity in employment life courses in Europe in the Twentieth Century—Large cross-national differences but little change across birth cohorts. Social Forces 96 (1): 1–30.

Overall, I think the paper suffers a bit from the use of many concepts. It would benefit from more explicitly focusing on the work psychology career literature. I would be more careful in generalizing to the labour market, even if the finding on the lack of career management in the most insecure positions can be seen as a contribution to labour market research also. In contrast to the framework used in the paper (e.g. p.11, row 32), I would argue that from the labour market segmentation theory point of view, it is not surprising that those in the most periphery positions suffer from the lack of career management, because they may have the lowest resources (e.g. achieved skills).

The paper uses high quality data and measurement, and overall, I suppose, the paper makes a contribution to job insecurity measurement, as well as to the theoretical development of work psychology career discussion.

Reviewer 2 Report

This is a well-focused paper that adresses significant research questions. Therefore is has a potential to make significant theoretical and empirical contibution to the job insecurity debate in relation to labour market segmentation and flexibilisation. For that to happen the paper needs some substantive improvements.

Firstly, the aims of the paper should be explicitly stated in the abstract and introduction. Secondly, and most importantly, the design needs to be improved. The study has used two volunteer samples to adress the questions about the distribution of job insecurity in population. Any kind of generalisations from non-probabilty samples like the samples used in this study are highly problematic and in this study even inappropriate. Especially when no justification is provided for why such sampling methods were chosen and how they affected sample composition and how these two samples compare to the general population of workers in Belgium or in Flanders. (For more on descriptive inferences from nonprobability samples see: Kohler, U., Kreuter, F., & Stuart, E., A. (2018). Nonprobability Sampling and Causal Analysis. Annual Review of Statistics and Its Application, 6(1), 149-172.) Unless the authors can provide a convincing justification for their sampling approach, I think their conclusions about the distribution of job insecurity profiles in the population are unjustified and the results cannot reliably be generalised..

The second part of the analyses (comparisions of employabliyt)  is less problematic but why only bivariate analysis is used? Both job insecurity and employability profiles are highly likely to be related to number of co-founding variables, such as occupation, education etc. Why at least some of the variables were not controlled for using appropriate regression analysis methods?

The paper is reasonably well written and the main arguments are very clear.
